# Tenderization of Beef *Semitendinosus* Muscle by Pulsed Electric Field Treatment with a Direct Contact Chamber and Its Impact on Proteolysis and Physicochemical Properties

**DOI:** 10.3390/foods12030430

**Published:** 2023-01-17

**Authors:** Se-Ho Jeong, Young-Min Jung, Siyeon Kim, Jong-Hun Kim, Hyunho Yeo, Dong-Un Lee

**Affiliations:** 1Department of Food Science & Technology, Chung-Ang University, Anseong 17546, Republic of Korea; 2R&D Center, Ottogi Ltd., Anyang 14060, Republic of Korea

**Keywords:** pulsed electric field (PEF), beef, meat quality, tenderization

## Abstract

In this study, the effects of pulse electric field (PEF) treatment on the tenderization of beef semitendinosus muscle were investigated. An adjustable PEF chamber was designed to make direct contact with the surface of the beef sample without water as the PEF-transmitting medium. PEF treatment was conducted with electric field strengths between 0.5 and 2.0 kV/cm. The pulse width and pulse number were fixed as 30 μs and 100 pulses, respectively. The impedance spectrum of PEF-treated beef indicated that PEF treatments induced structural changes in beef muscle, and the degree of the structural changes was dependent on the strength of the electric field. Cutting force, hardness, and chewiness were significantly decreased at 2.0 kV/cm (35, 37, and 34%, respectively) (*p* < 0.05). Troponin-T was more degraded by PEF treatment at 2.0 kV/cm intensity (being degraded by 90%). The fresh quality factors such as color and lipid oxidation were retained under a certain level of PEF intensity (1.0 kV/cm). These findings suggest that PEF treatment could tenderize beef texture while retaining its fresh quality.

## 1. Introduction

Pulsed electric field (PEF) treatment is a non-thermal technology for minimally processed food and is easy to apply continuously thanks to its low level of heat generation, short treatment time, and low energy consumption [1]. The use of PEF retains fresh food traits such as a texture, color, flavor, and nutrients compared with the use of thermal technologies [2]. The PEF treatment parameters are adjusted by electric field strength (kV/cm), pulse width (μs), pulse number (n), and pulse frequency (Hz) [3]. The field strength represents the intensity of PEF. The pulse width, number, and frequency determine the treatment time. PEF treatment induces an electrical breakdown of the cell membrane filled with cytoplasm, which is a dielectric medium [4].

There have been investigations of non-thermal technologies as a pretreatment for meat processing. High hydrostatic pressure (HPP) has been used to tenderize the texture of beef and to inactivate the surface microbial counts [5,6]; ultrasound treatment is effective for the brining of meats [7,8,9]; and cold plasma (CP) can inactivate the surface microbial contamination on meat and fish [10,11].

PEF has been reported to be effective in the tenderization of meat by decreasing the size of the muscle and enlarging the space between the muscle cells; thus, proteolysis is increased [12,13]. Furthermore, the pretreatment of meat with PEF increases the solubility of connective tissue during thermal processing by reducing the collagen denaturation temperature [14,15]. Studies have explored the impact of PEF on meat tenderness in post-rigor beef muscle [16], beef short ribs [17], and cold boned muscle [18]. Carne et al. [12] found that the pretreatment of beef with PEF could decrease the shear force by up to 21.6% depending on the intensity of the field and the frequency of the pulse. In contrast, O’Dowd et al. [16] reported that the instrumental texture of beef pretreated by PEF was unaffected. The tenderness of meat is influenced by the direction of meat cuts, the quantity of nutritional components, the variation in electrical characteristics, and the orientation of muscle fiber [17].

To date, the applications of PEF for meat products are very limited because water is used as the PEF-transmitting medium. The objective of this study was to verify the effects of PEF on fresh meat with an adjustable PEF chamber, which directly contacts the meat sample without water as the PEF-transmitting medium. The PEF effects on texture (tenderness), color, connective tissue protein, and the lipid oxidation of meat were assessed.

## 2. Materials and Methods

### 2.1. Sample Preparation and Reagents

Fresh beef (*Semitendinosus* muscle) within 48 h of slaughter was supplied from a local slaughterhouse (Anseong, Republic of Korea). The beef was preserved at 4 °C before slicing. The beef was sliced to a thickness of 10 mm along the longitudinal direction of the muscle fibers. The beef sample was cut into a rectangular shape of 2 × 2 × 1 cm after trimming unnecessary fats and tissues. All chemicals used were Sigma-Aldrich products (St. Louis, MO, USA). 

### 2.2. PEF Treatment

#### 2.2.1. Manufacture of the Adjustable PEF Chamber

Conventional PEF batch chambers are designed to treat food with water [19]. However, the blood and fat of meat can flow out during the immersion of meat. A new PEF chamber that could be used without water was designed to exclude any potential influence of water. Figure 1 shows the design of the adjustable PEF batch chamber that directly contacts the meat. The electrodes of the PEF chamber are made of titanium. Figure 1A shows that the electrodes are adjustable from 1 to 5 cm. 

#### 2.2.2. PEF Treatment of Beef 

This study used a pulse generator (HVP-5, DIL, Germany) with 5 kW of power. The PEF treatment was conducted in batch mode with the newly designed batch chamber. The electric field strength of the PEF treatment was adjusted using fixed parameters (pulse frequency: 20 Hz, pulse width: 30 μs, and pulse number: 100). The gap between the parallel stainless electrodes was adjusted to 1 cm. The sample was placed in the center of the electrodes and fixed by moving the electrodes. The power (%) of the generator was adjusted. The electrical field strength was calculated by the following equation:(1)$$\mathrm{Electrical}\;\mathrm{field}\;\mathrm{strength}\;\left(E,\;\mathrm{kV}/\mathrm{cm}\right)=\frac{\mathrm{Output}\;\mathrm{voltage}\;(\mathrm{kV})}{\mathrm{Distance}\;\mathrm{of}\;\mathrm{the}\;\mathrm{electrodes}\;\mathrm{gap}\;\left(\mathrm{cm}\right)\;}$$

The field strengths were 0.5, 1.0, 1.5, and 2.0 kV/cm and the corresponding power was 5, 10, 15, and 20%, respectively.

### 2.3. Estimation of Biological Electric Conductivity of PEF-Treated Beef

Cell permeability induced by PEF was estimated by measuring biological electric conductivity σ (S/m) at different frequencies. The electric conductivity of the beef sample was measured by LCR-8000G (Instek, New Taipei City, Taiwan) [20]. The equation for the electrical conductivity can be expressed as follows: (2)$$Electric\;conductivity\;\sigma \;\left(S/m\right)=L\times Y\times \frac{1}{A}$$
where L is the length of the sample, Y is the admittance of the sample, and A is the area of the sample [21].

### 2.4. Measurement of Texture Properties

Texture properties were measured as the cutting force and texture profile using TAHDi/500 (TAHD, Godalming, UK) with a blade probe (cutting force) and a P/35 cylindrical probe (TPA) [22,23]. All tests were conducted with the samples placed longitudinally along the muscle fiber. The cutting force was conducted by setting the speed at 4 mm/s and measuring max force. The texture profile analysis was conducted by setting a strain of 30% and a constant speed of 1 mm/s. All samples were a fixed size (cutting force: 2 × 1 × 1 cm, TPA: 2 × 2 × 1 cm) and temperature (4 °C). 

### 2.5. Troponin-T and μ-Calpain Quantification

Protein extraction was conducted using the following method [24]. Briefly, 5 g of beef sample was homogenized with 15 mL of radioimmunoprecipitation assay (RIPA) buffer at 13,500 rpm (2 °C, 30 seconds). The homogenate was centrifuged at 5500× g for 30 minutes at 4 °C. A total of 1 mL of the supernatant was transferred and centrifuged at 13,000× *g* (4 °C, 15 min). The sediment was washed twice with 15 mL of the buffer and centrifuged at 2300 × *g* (4 °C, 10 min). The solution was heated in a water bath (80 °C, 3 min) and the supernatant was stored after centrifugation at 13,000× *g* (4 °C, 15 min). The protein quantification was conducted using the bicinchoninic acid assay (BCA) method. Western blotting was conducted using the following methods with minor modifications [25,26]. Briefly, protein was loaded into SDS-PAGE and transferred to polyvinylidene difluoride (PVDF) membranes. The membranes were blocked in a buffer within 5% dried skim milk (*w*/*v*) for 1 h at room temperature (RT). Blots were incubated overnight at 4 °C, diluted to a 1:1000 ratio of troponin-T (mouse monoclonal anti-rabbit Troponin-T, clone JLT-12; T 6277, Sigma-Aldrich Corp., St. Louis, MO, USA) and μ-calpain antibodies (mousemonoclonal anti-bovine μ-calpain; MA3-940, Thermo Scientific, Rockford, IL, USA), respectively. All blots were incubated for 1 h at RT with the peroxidase-conjugated anti-mouse IgG (Sigma A 2554) antibody diluted to 1:3000, respectively. All bands were detected by chemiluminescence (ELC Plus^TM^, GE Healthcare, Buckinghamshire, UK). The band intensity was visualized by AlphaEaseFC^TM^ (Alpha innotech Corp., San Leandro, CA, USA). 

### 2.6. Color 

The color difference of the beef was expressed by *L*^*^ (lightness), *a*^*^ (redness (positive), greenness (negative)), and *b*^*^ (yellowness (positive), blueness (negative)) as CIELab scales and measured by UltraScan Pro (HunterLab, Reston, VA, USA). The color difference between control and treated samples is expressed as Δ*E*. Δ*E* was calculated using the following equation:(3)$$∆E=\sqrt{{\left(∆L^{*}\right)}^{2}+{\left(∆a^{*}\right)}^{2}+{\left(∆b^{*}\right)}^{2}}$$

### 2.7. Measurement of Lipid Oxidation

Lipid oxidation was evaluated by thiobarbituric acid reactive substances (TBARs) and expressed as mg MDA/kg meat. The TBARs test was conducted using the following method [27]. Briefly, 5g of sample was homogenized with 15 mL of distilled water and centrifuged at 2000× *g* for 15 min. The supernatant was filtrated. A total of 1 mL of the supernatant was transferred with 2 mL of trichloroacetic acid/thiobarbituric acid (TCA/TBA) solution (15% TCA (*w*/*v*), 0.375% TBA (*w*/*v*) in 0.25 M HCl) and 3 mL of butylated hydroxytoluene (BHT) solution (2% BHT (*w*/*v*) in 99% ethanol). The mixture was heated at 95 °C for 15 minutes and rapidly cooled down. The mixture was centrifuged at 1000× *g* for 10 min. The supernatant was transferred and absorbance measured at 530 nm. The blank test was conducted using distilled water and a standard solution was used with 1,1,3,3-tetraethoxypropane.

### 2.8. Statistical Analysis

All experiments were conducted in six parallel measurements (*n* = 6). All data are displayed as mean ± standard deviation. The significance was determined by one-way ANOVA at the level of *p* < 0.05 using the IBM SPSS ver. 20 (IBM Corp., Chicago, IL, USA). The difference between each sample was determined by Duncan’s multiple range test at the level of *p* < 0.05.

## 3. Results and Discussion

### 3.1. Electrical Conductivity Spectrum of PEF-Treated Beef 

Figure 2 shows the electrical conductivity spectra of beef samples. The electrical conductivity was proportionally increased by the field strength. The control and 0.5 kV/cm-treated beef sample had a similar conductivity spectrum. The 2.0 kV/cm-treated beef sample showed the highest conductivity at all frequencies. Similar to the present study, it was reported that the conductivity of meat was increased by PEF treatment [13,28]. The increased conductivity in muscle enhanced the drip that flowed out between muscle fibers [29]. The drip could occupy the spaces between muscle fiber [30]. This phenomenon enhanced the flow of current in muscle fiber owing to enhanced ion transport. The increasing conductivity in meat shows a change in the membrane permeability related to drip loss [31]. These results indicate that PEF treatment induces an increase in the permeability of muscle fiber.

### 3.2. Texture Properties of PEF-Treated Beef

#### 3.2.1. Cutting Force

Figure 3 shows the cutting force/shear force of the beef sample according to PEF treatment. The texture evaluation of beef was expressed by the value of the cutting force (N). The cutting force of the control sample was measured to be 67.74 ± 9.27 N, and that at 0.5 kV/cm and 1.0 kV/cm was measured to be 60.09 ± 4.36 N and 53.91 ± 6.15 N, respectively. The cutting force of the 1.5 kV/cm- and 2.0 kV/cm-treated samples was measured to be 48.34 ± 2.65 N and 39.57 ± 1.76 N, respectively. The cutting force of all treated samples was significantly different from the control and decreased in proportion to the field strength. These results may have been caused by the degradation of the myofibril structure. The space between the muscle fiber bundle and perimysium was expanded by PEF treatment [16]. In addition, post-PEF treatment, a transmission electron micrograph (TEM) of the beef muscle showed ruptures along the Z-lines, unlike the intact muscle [28]. Section 3.1. shows that PEF treatment caused spaces between muscle fibers. These spaces, occupied by the drip, could weaken the binding between muscle fiber bundles. PEF treatment has an effect on proteolysis during post-mortem storage, increasing the degradation of troponin-T and desmin [32]. Similar to the present study, PEF treatment significantly reduced the shear force of beef longissimus thoracis et lumborum muscle [23]. It is thus likely that PEF treatment can tenderize beef muscle. 

#### 3.2.2. Texture Profile Analysis (TPA)

Table 1 shows the texture profiles of the control and PEF-treated beef samples. The hardness of treated beef at 2.0 kV/cm was significantly reduced from 4.05 ± 0.65 to 2.35 ± 0.38 N (*p* < 0.05). However, there was no significant difference between the PEF-treated groups. The PEF treatment reduced beef chewiness from 2.60 ± 0.27 to 1.65 ± 0.26 (*p* < 0.05). The cohesiveness was slightly increased as the strength of PEF increased (*p* < 0.05). Springiness was not affected by PEF treatment. Although the present study specifically observed the texture profile of raw beef, the cooked beef post PEF treatment showed tenderization. It was reported that PEF pretreatment was maintained after sous vide cooking [33,34]. The texture profile of PEF-treated raw beef has not been studied in depth; however, the PEF treatment resulted in a decrease in the hardness and chewiness of plants, as in the present study. Plants with tough textures such as apple, potato, and ginseng were softened by PEF treatment [20,35]. Thus, it is suggested that the application of PEF treatment to beef contributes to tenderization.

### 3.3. Proteolysis of PEF-Treated Beef

Figure 4 shows the relative quantification of the Western blot for beef processed with PEF at different field strengths. Troponin-T is an indicator protein in muscle contraction that binds to tropomyosin. When troponin-T is degraded, the contracted muscle is relaxed [36]. The amount of troponin-T can be relatively quantified using μ-calpain. μ-Calpain is a protein-hydrolytic enzyme that exists only in animal cells. The reason for using μ-calpain is that it is not degraded by PEF treatment. The amount of control at 1.0 kV/cm was approximately 0.32 and that at 2.0 kV/cm was approximately 0.10. The decreased relative quantity of troponin-T complements the tenderization phenomenon shown by cutting force and TPA. Similar to the present study, PEF treatment affects the rigor mortis stage via the degradation of troponin-T and desmin [13]. These results demonstrate that PEF treatment affects tenderization.

### 3.4. Color Change in Treated Beef

Table 2 shows the color of beef samples according to PEF treatment. The lightness (*L*^*^ value) of the control beef was 42.52 ± 1.11, which was 2.0 kV and slightly decreased to 40.07 ± 0.54 after PEF treatment. However, PEF treatment of 1.0 kV or higher did not show a significant difference in the value of *L*^*^, *a*^*^, and *b*^*^ (*p* < 0.05). Redness (*a*^*^) and yellowness (*b*^*^) showed no difference before and after the PEF treatment of beef. This result indicates that PEF treatment has the least effect on the color of beef, which can be confirmed by small ΔE values. In particular, the *a*^*^ value, which is the dominant color of beef, is not changed. According to the research by O’Dowd et al., the PEF-treated beef exhibited a slightly decreased *L*^*^ value (*p* < 0.05) and retained its *a*^*^ and *b*^*^ value (*p* > 0.05) [16]. The research on PEF-treated pork shows slightly decreased lightness with increased field strength [37]. The color of lamb is not significantly influenced by PEF [38]. Thus, the color difference in PEF-treated beef was mechanically small; however, it was in the range that was not visually perceived.

### 3.5. Lipid Oxidation of PEF-Treated Beef

Figure 5 shows the lipid oxidation of PEF-processed beef at different field strengths. As a result, the TBARs of the control were measured as 1.10 ± 0.06 mgMDA/kg, and those of the beef treated with 2.0 kV/cm PEF were measured as 1.36 ± 0.05 mgMDA/kg. The PEF treatment slightly increased the value of TBARs to the level of 0.5 mgMDA/kg compared with the control. Similar to these results, a previous report suggested that lipid oxidation was accelerated when thawed beef was treated with PEF [39]. Additionally, it was confirmed that the difference between the TBARs of the control and 2.0 kV/cm PEF-treated beef was less than 0.3 mgMDA/kg. In the case of HPP treatment, the lipid oxidation in the muscle is caused by the release of metal ions from the iron complex, accelerating auto-lipid oxidation [40]. PEF treatment increased the metal ion transport in muscle fiber bundles, and the high-voltage pulse promoted auto-oxidation. These results suggest that PEF treatment should be conducted at the correct level to minimize lipid oxidation.

## 4. Conclusions

In conclusion, the present study confirmed the effect of PEF treatment on the tenderization of semitendinosus muscle. The adjustable PEF batch chamber designed for this work excluded the influence of water. The chamber could be used to observe the effect of PEF by adjusting the electrode gap and fixing the treatment direction. The tenderization as a result of PEF treatment was confirmed through mechanical measurement and Western blot. The color of the beef was slightly changed by PEF treatment. However, the redness of the beef was retained. Although PEF treatment increased lipid oxidation under the present conditions, this may be overcome by adjusting detailed treatment conditions such as field strength. Thus, PEF treatment can tenderize the beef texture while retaining its fresh quality. However, our study was conducted under limited conditions because of the chamber size and the generator power. Subsequent research should focus on industrial applications, including larger samples, higher field strength, and higher productivity.

## Figures and Tables

**Figure 1 foods-12-00430-f001:**
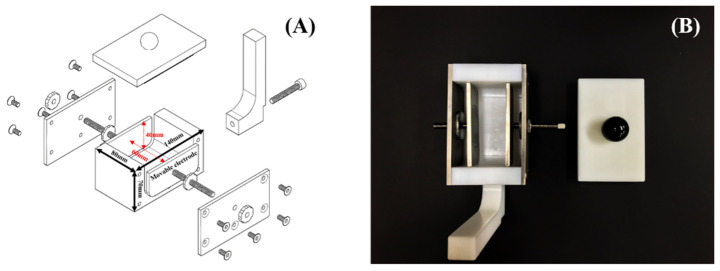
Models of the adjustable PEF batch chamber. (**A**) Three-dimensional assembly drawings, (**B**) final shape of the chamber.

**Figure 2 foods-12-00430-f002:**
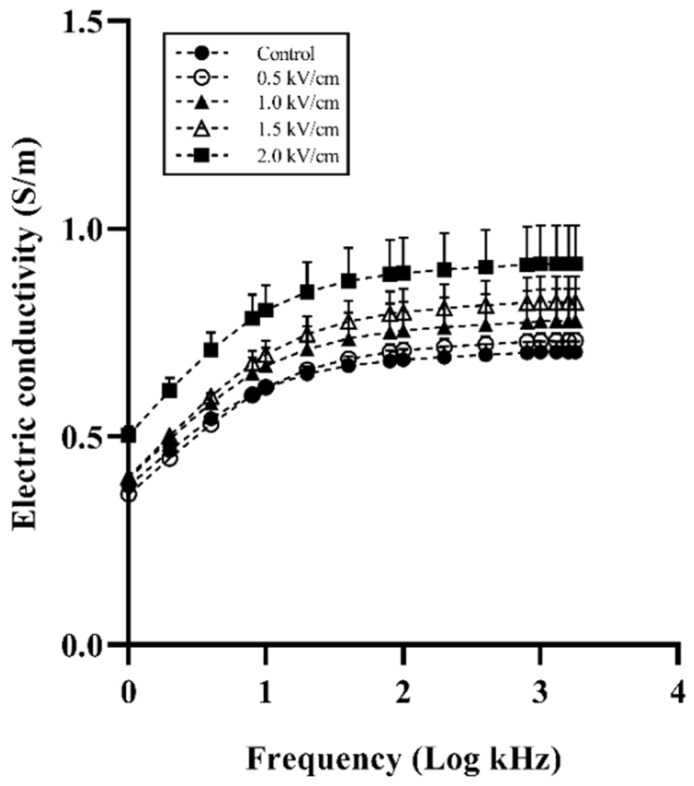
Electrical conductivity spectra of beef with different field strengths. Error bars indicate the standard deviation of six replications (*n* = 6).

**Figure 3 foods-12-00430-f003:**
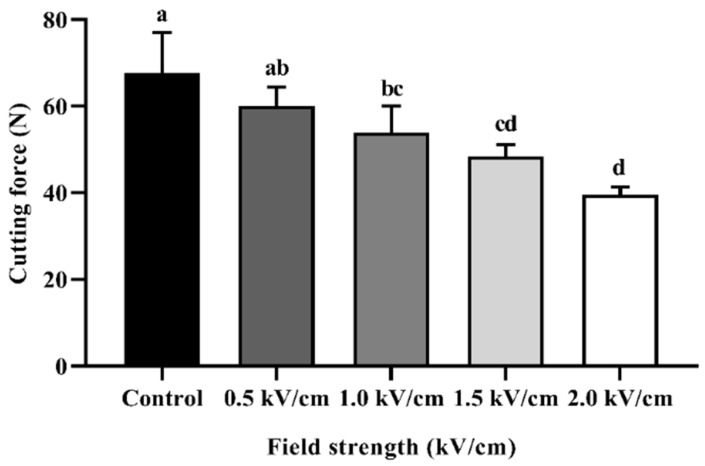
Cutting force for beef processed with PEF at different field strengths. Error bars indicate the standard deviation of six replications (*n* = 6). Different letters indicate significant differences between means (*p* < 0.05).

**Figure 4 foods-12-00430-f004:**
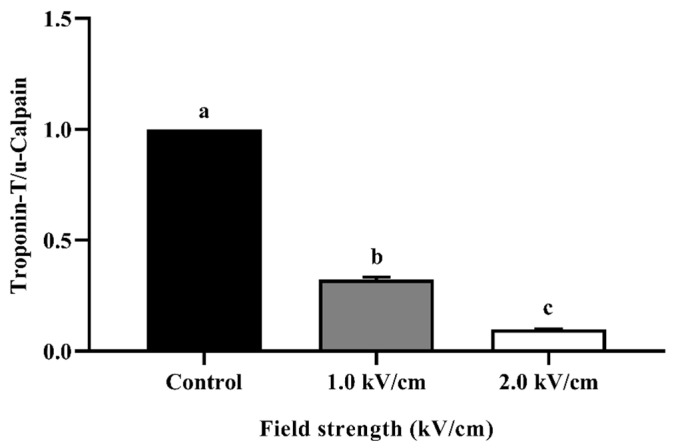
Relative quantification of Western blot for PEF-treated beef with different field strengths. Error bars indicate the standard deviation of six replications (*n* = 6). Different letters indicate a significant difference (*p* < 0.05).

**Figure 5 foods-12-00430-f005:**
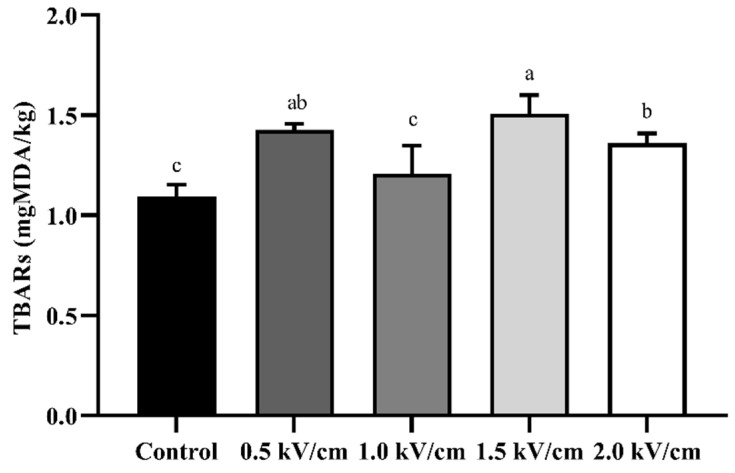
Lipid oxidation using TBARs for beef processed with PEF at different field strengths. Error bars indicate the standard deviation of six replications (n = 6). Different letters indicate a significant difference (*p* < 0.05).

**Table 1 foods-12-00430-t001:** Texture profile analysis (TPA) of PEF-treated beef at different field strengths.

Sample	Hardness (N)	Springiness	Cohesiveness	Chewiness (N)	Resilience
Control	4.05 ± 0.65 ^a^	0.95 ± 0.02 ^a^	0.68 ± 0.03 ^a^	2.60 ± 0.27 ^a^	0.23 ± 0.02 ^a^
PEF with 0.5 kV/cm	2.98 ± 0.18 ^b^	0.97 ± 0.01 ^a^	0.71 ± 0.01 ^a^	2.03 ± 0.15 ^b^	0.20 ± 0.04 ^a^
PEF with 1.0 kV/cm	2.72 ± 0.22 ^b^	0.96 ± 0.01 ^a^	0.70 ± 0.02 ^ab^	1.82 ± 0.09 ^bc^	0.18 ± 0.02 ^a^
PEF with 1.5 kV/cm	2.51 ± 0.28 ^b^	0.97 ± 0.01 ^a^	0.73 ± 0.02 ^ab^	1.77 ± 0.15 ^bc^	0.21 ± 0.04 ^a^
PEF with 2.0 kV/cm	2.35 ± 0.38 ^b^	0.96 ± 0.01 ^a^	0.73 ± 0.01 ^b^	1.65 ± 0.26 ^c^	0.18 ± 0.00 ^a^

All values are expressed as the mean ± standard deviation (*n* = 6). Means with different letters in the same column are significantly different (*p* < 0.05).

**Table 2 foods-12-00430-t002:** Color measurement of beef processed with PEF at different field strengths.

Sample	*L**	*a**	*b**	Δ*E*
Control	42.52 ± 1.11 ^a^	17.67 ± 2.52 ^a^	11.92 ± 1.48 ^a^	-
PEF with 0.5 kV/cm	41.83 ± 0.62 ^a^	19.59 ± 0.27 ^a^	13.00 ± 0.48 ^a^	2.40 ± 0.18
PEF with 1.0 kV/cm	40.15 ± 0.47 ^b^	18.00 ± 3.17 ^a^	11.71 ± 1.25 ^a^	3.68 ± 0.41
PEF with 1.5 kV/cm	39.89 ± 0.15 ^b^	18.66 ± 0.87 ^a^	12.47 ± 0.67 ^a^	3.00 ± 0.21
PEF with 2.0 kV/cm	40.07 ± 0.54 ^b^	17.78 ± 1.35 ^a^	11.95 ± 0.30 ^a^	2.68 ± 0.66

All values are expressed as the mean ± standard deviation (n = 6). Means with different letters in the same column are significantly different (*p* < 0.05).‘-’ means ‘no value’.

## Data Availability

Data is contained within the article.

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
