# Peer review of "Tenderization of Beef Semitendinosus Muscle by Pulsed Electric Field Treatment with a Direct Contact Chamber and Its Impact on Proteolysis and Physicochemical Properties"

_foods, 2023, doi:10.3390/foods12030430_

Round 1
Reviewer 1 Report
The manuscript ‘Tenderization of Beef Semitendinosus Muscle by Pulsed Electric Fields Treatment and Its Impact on Proteolysis and Physico-chemical Properties’ present some interesting findings. The hypothesis is well explained and clear. The language of the manuscript is clear and easy to understand. The experimental design of an adjustable PEF batch chamber using it without water is novel in this study.
I have following suggestions-
Abstract:
i. L 16; plz mention p value
ii. L19: please mention the intensity
Keywords: replace either texture or tenderness with meat quality
Introduction: well, outlined the sufficient background and need for carrying out the study.
iii. L33-35: Authors may be focused on PEF, may compare it with other novel processing technologies.
iv. L 55: tenderization of texture; plz check the word for improving tenderization and texture ----
i. L67: All chemicals used were
ii. L180: cutting force/ shear force
iii. L243: plz check sentence
iv. L269-270- The PEF treatment-----------MDA/kg; please check it and rewrite
Author Response
Dear Reviewer 1,
Authors are grateful to the editor and reviewer for interest in findings and valuable suggestions that really helped in the improvement of the manuscript. A point-by-point response to the editor’s comments is included below. All changes in the revised manuscript are highlighted in red text font.
Reviewer’s comments:    
|
No. |
question |
line |
→ |
answer |
line |
|
1 |
plz mention p value
|
16 |
We have added this information
“ Cutting force, hardness, and chewiness were significant decreased at 2.0 kV/cm (35%, 37%, and 34% respectively) (p < 0.05).” |
20 |
|
|
2 |
please mention the intensity
|
19 |
We have added this information
“ The fresh quality factors such as color and lipid oxidation were retained under a certain level of PEF intensity (1.0 kV/cm).” |
22 |
|
|
3 |
Keywords: replace either texture or tenderness with meat quality |
21 |
We have corrected this information
“Keywords: pulsed electric field (PEF), beef, meat quality, tenderization” |
24 |
|
|
4 |
Authors may be focused on PEF, may compare it with other novel processing |
33-35 |
We revised this part
“There have been ~ muscle fibre |
37-53 |
|
|
5 |
tenderization of texture; plz check the word for improving tenderization and |
55 |
We deleted this part for rephrase more accurately. |
54-58 |
|
|
6 |
All chemicals used were |
67 |
We have corrected this information
“All chemicals used were Sigma-Aldrich products (St. Louis, MO, USA).” |
65 |
|
|
7 |
cutting force/ shear force |
180 |
We have corrected this information
“ Figure 3 shows the cutting force/shear force of the beef sample according to PEF treatment. ” |
184 |
|
|
8 |
plz check sentence |
243 |
We revised this part
“ However, PEF treatment of 1.0 kV or higher did not show a significant difference in the value of L*, a* and b* (p < 0.05). ” |
250 |
|
|
9 |
The PEF treatment-----------MDA/kg; please check it and rewrite |
269-270 |
We revised this part
“ The PEF treatment slightly increased the value of TBARs to the level of 0.5 mgMDA/kg compared with the control .” |
275 |

Author Response
Dear Reviewer 2
Authors are grateful to the editor and reviewers for interest in findings and valuable suggestions that really helped in the improvement of the manuscript. A point-by-point response to the editor’s comments is included below. We also checked the English of our manuscript by a professional English editor suggested by Foods. All changes in the revised manuscript are highlighted in red text font.
Reviewer’s comments:    
|
No. |
question |
line |
→ |
answer |
line |
|
1 |
This is interesting technology which I think could have some application in small batch systems. I do not know what the authors have done relative to experimental design/replication and do not feel I can adequately assess the validity of the technology. I do not know how many measurements went into generating each mean. Were the samples cooked? If so, how? How many repetitions? Include more detail of methods for textural analysis
|
|
Our experiments were conducted in six parallel measurements (n = 6). We add the replication of each experiment on the table and figure legends.
All samples were a fixed size (cutting force: 2 × 1 × 1 cm, TPA: 2 × 2 × 1 cm) and tem-perature (4°C). We revised methods part |
112
|

Reviewer 3 Report
Title: the novelty or innovation of the research could be challenged because there is extensive research in this field has been already published (a few of them is mentioned below):
1. https://doi.org/10.1007/s11947-014-1324-8
2. https://doi.org/10.1016/j.meatsci.2012.09.010
3. https://doi.org/10.1016/j.meatsci.2015.02.009
4. https://doi.org/10.3390/foods11182803
5. https://doi.org/10.1016/j.meatsci.2014.10.011
Abstract: should include the background to the contribution of your paper, the main objectives of your research, and at the end the impact of your research.
Introduction: Introduction has a good length but is very poor in language and structure. Each sentence needs to be rephrased for better understanding. The background of research and review of the literature makes no sense or connections with the relevant research title. Authors must discuss previous literature and similar topics on beef and tenderization with PEF and must justify why their contribution is needed. The details of the studies are also poorly constructed.
Material and methods: Poorly described with a lot of grammatical errors and technical confusion. Insufficient detail is provided about the PEF procedure and the parameters used for PEF. Inconsistent presentation of PEF parameters in the manuscript. Moreover, the authors indicated that the meat samples were treated in PEf without water. Then how the uniform electrolysis could be possible? Because water is used as a medium to convey electricity. The authors have not conducted pH measurements, which is the most important test to estimate beef muscle stages (Rigor Mortis) and is also directly linked to the textural properties and tenderness of meat muscles.
Results and discussion: Since the effect of different PEF treatments is discussed, these treatments have to be better characterized in terms of field strength, treatment time, and specific energy. Also when compared with the publications of other authors. Sometimes field strength and the number of pulses are only indicated which is not sufficient to characterize the PEF treatments and to compare results. The inclusion of all PEF parameters will give better support for the discussion of the influence of the investigated parameters.
Conclusion: It is customary to mention future work, for example, to solve some of the limitations stated in the paper.
English language: Although, I am not a native English speaker, the manuscript requires extensive English correction, and rephrasing. The document contains various typographical and grammatical errors, that require careful review by a fluent English speaker - and many inconsistencies (some of which I have highlighted in the attached file).
Please see the attached file for more comments.

Author Response
Dear Reviewer 3,
Thank you very much for your detailed- and valuable comments on our manuscript. Authors are really grateful to the reviewer for interest in findings and valuable suggestions that really helped in the improvement of the manuscript.
-
The title, abstract, and introduction were extensively revised based on the reviewer's suggestions. We also cite and discuss the references pointed out by reviewer.
- We also checked the English of our manuscript by a professional English editor suggested by Foods.
- A point-by-point response to the reviewer’s comments is included below. All changes in the revised manuscript are highlighted in red text font.
Reviewer’s comments:    
|
No. |
question |
line |
→ |
answer |
line |
|
1 |
PEF treated, Pulse
|
14 |
We have added this information “PEF-treated, pulses”
|
16, 17 |
|
|
2 |
The impedance spectrum of PEF treated beef indicated that PEF treatments induced structural changes of beef muscle, and the degree of structural changes were electric field strength dependent. |
15 |
We have added this information The impedance spectrum of PEF-treated beef indicated that PEF treatments induced structural changes in beef muscle, and the degree of structural changes was dependent on the strength of the electric field.”
|
16 |
|
|
3 |
Certain level? Which level?
The fresh quality factors such as color and lipid oxidation were retained under a certain level of PEF intensity.
These findings suggest that PEF treatment could tenderize beef texture retaining their fresh quality. |
19-20 |
We have corrected this information
“ The fresh quality factors such as color and lipid oxidation were retained under a certain level of PEF intensity (1.0 kV/cm). ”
“These findings suggest that PEF treatment could tenderize beef texture while retaining its fresh quality..” |
22-23 |
|
|
4 |
do you mean nonthermal technology?
minimally processed
nutrients |
25-28 |
We revised this part more accurately.
“Pulsed electric field ~ technologies.” |
28-31 |
|
|
5 |
Too many sites to revise, so we pointed with red color in files. |
32-60 |
We rewrite these parts. “There have been~ were assessed |
37-58 |
|
|
6 |
The word after seems to be unintentionally repeated |
66 |
We have corrected this part.
|
64 |
|
|
7 |
within |
72 |
with
|
70 |
|
|
8 |
Fixed parameters
on which basis do you select these parameters? is there any previous study or trial to select these parameters? |
86 |
The effect of PEF was depended by an electric field strength or a treatment time. In this study, we have fixed the treatment time that is set by adjusting frequency, pulse number, and pulse width. The field strength is depended on an electric conductivity of sample and power (%) of pulse generator. Meat has a high content of fat and protein. It can be denatured by heat during the PEF treatment. Thus, we have found a conditions with low heat generation through preliminary experiments. In our previous results, the heat generation was affected by treatment time than field strength. |
|
|
|
9 |
admittance of samples what is it? |
96 |
The admittance means how easily the sample will allow a current. The unit of admittance is a siemens(S). The LCR device measures the admittance of the samples. We have converted the admittance(S) to an electric conductivity(S/m)
|
|
|
|
10 |
The cutting force was conducted setting a constant speed at 4 109 mm/s and measured max force. The TPA was conducted setting a strain of 30% and a 110 constant speed of 1 mm/s.
by setting, add full name first then abbreviate it . |
109-110 |
|
There are slight grammar issue thus we corrected this part that you pointed. Thank you. |
112-113 |
|
11 |
Protein extraction was conducted using a followed method . |
114 |
|
There are slight grammar issue thus we corrected this part that you pointed. Thank you. |
|
|
12 |
were |
123 |
|
There are slight grammar issue thus we corrected this part that you pointed. Thank you. |
|
|
13 |
Oxidation is generally singular and uncountable |
143 |
|
There are slight grammar issue thus we corrected this part that you pointed. Thank you. |
|
|
14 |
The supernatant was transferred and measured an absorbance at 530 nm. Blank test was conducted by distilled water and a standard solution was used by 1,1,3,3- tetraethoxypropane. |
152 |
|
There are slight grammar issue thus we corrected this part that you pointed. Thank you. |
|
|
15 |
Electrical, a, drip loss |
164-173 |
|
There are slight grammar issue thus we corrected this part that you pointed. Thank you. |
|
|
16 |
Control, were, statistically significant, muscle, were, show, fiber |
183-191 |
|
There are slight grammar issue thus we corrected this part that you pointed. Thank you. |
|
|
17 |
rephrase it |
207 |
|
|
|
|
18 |
Difference, the, texture, potato and, the application |
208-214 |
|
There are slight grammar issue thus we corrected this part that you pointed. Thank you. |
|
|
19 |
Strength, on, to, has an effect on |
224-236 |
|
There are slight grammar issue thus we corrected this part that you pointed. Thank you. |
|
|
20 |
Especially, the, of, difference, however |
248-254 |
|
There are slight grammar issue thus we corrected this part that you pointed. Thank you. |
|
|
21 |
the |
278 |
|
There are slight grammar issue thus we corrected this part that you pointed. Thank you. |
|
|
22 |
To exclude, clearly, gab |
287-288 |
|
There are slight grammar issue thus we corrected this part that you pointed. Thank you. |
|
|
23 |
Insufficient detail is provided about the PEF procedure and the parameters used for PEF. Inconsistent presentation of PEF parameters in the manuscript |
|
|
We add the details of PEF procedure.
The details of parameters were added in ‘introduction’ part
|
86 |
|
24 |
Moreover, the authors indicated that the meat samples were treated in PEF without water. Then how the uniform electrolysis could be possible? Because water is used as a medium to convey electricity. The authors have not conducted pH measurements, which is the most important test to estimate beef muscle stages (Rigor Mortis) and is also directly linked to the textural properties and tenderness of meat muscles. |
|
|
Thanks for your opinion. The PEF was induced irreversible cell permeability that was heterogeneous. With or without water, the PEF did not equally affect all cells because all samples were not uniform structure. The number of treated cells merely increased with the field strength and treatment time. We had conducted pH measurement that did not include manuscript. We attach a supplement table. Briefly, the pH of Control and treated sample were not significant differences (p<0.05). |
|
|
25 |
Since the effect of different PEF treatments is discussed, these treatments have to be better characterized in terms of field strength, treatment time, and specific energy. Also when compared with the publications of other authors. Sometimes field strength and the number of pulses are only indicated which is not sufficient to characterize the PEF treatments and to compare results. The inclusion of all PEF parameters will give better support for the discussion of the influence of the investigated parameters. |
|
|
Generally, the specific energy of each PEF treatment is proportional to field strength. In case of this study, we have fixed treatment time because of increasing sample temperature. Some studies are not sufficient to characterize the PEF treatments by the field strength and treatment time because these studies mostly deal with big size sample or low field strength. However, our study was performed using a small unit size and a high field strength. It is therefore that our results may be a sufficient explain. In subsequent study, we need the inclusion of all parameters for application in hard conditions like your opinion. |
|
|
26 |
It is customary to mention future work, for example, to solve some of the limitations stated in the paper. |
|
|
We have revised ‘conclusion’ part. |
|
Supplement table
Table S1. Change of pH and temperature for beef processed with PEF at different field strengths. Control, PEF treatments at 0.5 kV/cm, 1.0 kV/cm, 1.5 kV/cm and 2.0 kV/cm with a pulse width of 30μs, pulse number of 100.
|
  |
pH |
Temperature (℃) |
|
Control |
5.40 ± 0.01 |
19.3 ± 0.15e |
|
PEF with 0.5 kV/cm |
5.36 ± 0.01 |
22.2 ± 0.46d |
|
PEF with 1.0 kV/cm |
5.32 ± 0.02 |
23.7 ± 0.26c |
|
PEF with 1.5 kV/cm |
5.31 ± 0.01 |
26.2 ± 0.15b |
|
PEF with 2.0 kV/cm |
5.32 ± 0.01 |
29.4 ± 0.20a |

Round 2
Reviewer 3 Report
The overall manuscript is written well and brought some conclusions at the end of the manuscript. In the present manuscript, the authors have made an excellent effort to address all of the issues raised in the previous manuscript. Thank you.